# Proglacial methane emissions driven by meltwater and groundwater flushing in a high Arctic glacial catchment

Gabrielle E. Kleber[1,2,3], Leonard Magerl[3], Alexandra V. Turchyn[3], Stefan Schloemer[4], Mark Trimmer[5], Yizhu Zhu[5], Andrew Hodson[2,6]

[1]Department of Earth Sciences, University of Cambridge, Cambridge CB2 3EQ, UK
[2]Arctic Geology, University Centre in Svalbard (UNIS), Longyearbyen, 9170 Norway
[3]Department of Geoscience, UiT the Arctic University of Norway, Tromsø, 9010 Norway
[4]BGR – Federal Institute for Geosciences and Natural Resources, Hannover, 30655 Germany
[5]School of Biological and Behavioural Sciences, Queen Mary University of London E1 4NS, London, UK
[6]Department of Environmental Sciences, Western Norway University of Applied Sciences, Sogndal, 6856 Norway

*Correspondence to*: Gabrielle E. Kleber (gakle2914@uit.no)

**Abstract.** Glacial groundwater is a conduit for geologic methane release in areas of glacier retreat on Svalbard, representing a large, climate-sensitive source of the greenhouse gas. Methane emissions from glacial melt rivers are known to occur in other regions of the Arctic, but such emissions have not yet been considered on Svalbard. Over the summer of 2021, we monitored methane concentrations in the proglacial groundwater springs and river network of a ~20 km$^2$ valley glacier in central Svalbard to estimate melt season emissions from a single catchment. We measured methane concentrations in the glacial river up to 3170 nM (nearly 800-times higher than the atmospheric equilibrial concentration) and found the methane to be of thermogenic origin through isotopic analysis. We estimated a total of 1.0 ton of methane emissions during the 2021 melt season from the catchment, of which nearly two-thirds are being flushed from the glacier bed by the melt river. These findings provide further evidence that terrestrial glacier forefields on Svalbard are hotspots for methane emissions, with a climate feedback loop driven by glacier melt. As the first investigation into methane emissions from glacial melt rivers on Svalbard, our study suggests that summer meltwater flushing of methane from beneath the ~1400 land-terminating glaciers across Svalbard may represent an important seasonal source of emissions. Glacial melt rivers, including those from small valley glaciers, may be a growing emission point for subglacial methane across other rapidly warming regions of the Arctic.

## 1 Introduction

Components of the Arctic carbon cycle that naturally emit methane, such as wetlands, permafrost and geological seeps, are sensitive to climatic and seasonal changes (McGuire et al., 2009; Schuur et al., 2015; Walter Anthony et al., 2012; Yvon-Durocher et al., 2014; Zona et al., 2016). The vulnerability of these climate-sensitive systems to rising temperatures and changes in seasonal patterns have been reflected in increased methane emissions. This has led researchers to predict further increases in natural methane emissions from across the Arctic as global temperatures continue to rise—a positive feedback

that contributes to the amplification of warming in the Arctic and may increase the rate of future climate change (Schuur et al., 2008).

The Arctic hosts a large reservoir of organic carbon (Gautier et al., 2009; Hugelius et al., 2014; Isaksen et al., 2011; Wadham et al., 2019) which is stored in permafrost, natural gas deposits and coal beds. With sufficiently low temperatures and high pressures, volatile compounds like ethane and methane can be stored in a solid state in the form of gas hydrates. These subsurface carbon stores can be released to the atmosphere as methane gas, primarily by the microbial degradation of organic carbon once it becomes bioavailable via permafrost thaw or, alternatively, by the dissociation of gas hydrates and their direct release in response to climate warming. A growing body of research has identified additional pathways for natural methane emissions at the boundaries of glacial retreat in the Arctic, where active releases of both microbially-produced and geologic methane have been found to exist (Christiansen and Jørgensen, 2018; Kleber et al., 2023; Lamarche-Gagnon et al., 2019; Walter Anthony et al., 2012).

The advance of glaciers over vegetation secures a subglacial reservoir of organic carbon that can be microbially degraded into methane, which is then trapped by the overburden of the overlying glacier and accumulates (Vinšová et al., 2022; Wadham et al., 2019). Studies have detected methane releases at margins of retreating ice sheets and glaciers in Canada, Greenland and Iceland, where microbially-produced methane in the anoxic environment of the glacier bed is transported by meltwater and degassed to the atmosphere (Burns et al., 2018; Christiansen et al., 2021; Christiansen and Jørgensen, 2018; Dieser et al., 2014; Lamarche-Gagnon et al., 2019; Pain et al., 2020; Sapper et al., 2023). The findings of Lamarche-Gagnon et al (2019) suggest that the methane reserves beneath the Greenland Ice Sheet greatly exceed the methane transported to its margin, and thus increased melt in the future may lead to increased export and release of methane.

Alternatively, studies in Alaska and the Norwegian high Arctic have identified climate-sensitive releases of fossil geologic methane. In regions of permafrost thaw and glacier retreat, methane that was previously stored within rocks and trapped beneath a 'cryospheric cap' of glaciers and permafrost is now migrating to the surface and being released to the atmosphere (Kleber et al., 2023; Walter Anthony et al., 2012). Over 100 of these seeps have been identified across a region of Svalbard, Norway, where methane is brought to the surface by groundwater springs that form in the exposed forefields of retreating glaciers (Kleber et al., 2023). Emissions from these sources are expected to increase as more land is exposed by accelerating glacier melt.

The seasonality, extent, and the governing mechanisms of climate-enhanced methane emissions in the Arctic are still largely unknown and thus difficult to quantify. To gain a better understanding of methane emission dynamics in the climate-sensitive glacial environment, we have monitored the various methane sources found in the catchment of a single valley glacier on Svalbard. We have taken frequent water samples of the glacial melt river and groundwater springs to observe how their

methane content varied throughout the course of a melt season. Furthermore, we have estimated the potential melt-season methane emissions from these sources and addressed the origin of the methane while proposing a mechanism for its mobilization. Our study presents the first investigation into methane emissions associated to glacial melt rivers on Svalbard.

## 2 Methods

### 2.1 Site description and field study

Our study was based in the 42 km$^2$ hydrological catchment of the Vallåkrabreen glacier, a ~20 km$^2$ valley glacier located in central Svalbard (Fig. 1a). Vallåkrabreen is situated most prominently within the Carolinefjellet geological formation, a lithostratigraphic unit comprised of Lower Cretaceous organic-rich successions of fine-grained shales and sandstones. The Carolinefjellet Formation is a known petroleum source rock with inclusions of oil-associated thermogenic $C_1$-$C_4$ gases, possibly migrated upwards from lower Jurassic shale formations (Abay et al., 2017). The main field study took place between

July-September of 2021, with additional samples taken in 2022 and 2023. We measured methane concentrations in the glacial melt river and groundwater streams to estimate potential melt season methane emissions due to degassing. In addition, we measured methane ebullition (bubbling of gas) from vents within groundwater pools.

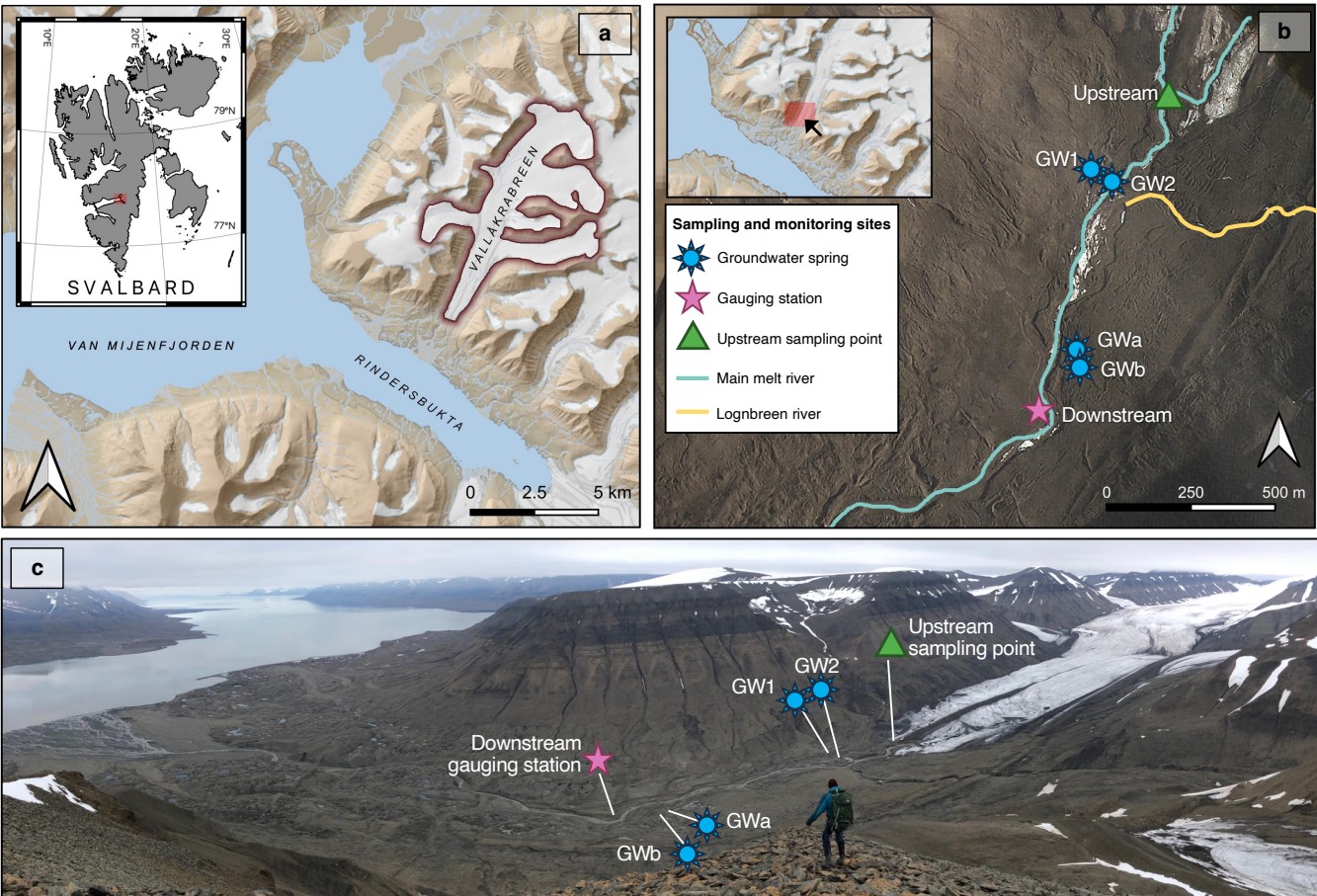

Figure 1. Overview of the Vallåkrabreen catchment. (a) Location of Vallåkrabreen on the Svalbard archipelago (base map data provided by the Norwegian Polar Institute), (b) location of sampling sites and the gauging station (satellite image retrieved on 07 July 2022 by Kompsat-2), (c) photo of the Vallåkrabreen catchment, taken from the location marked by the black arrow on the inset map in panel b.

We took water samples to measure methane concentrations in the glacial melt river every 2-5 days during the summer of 2021 at an 'upstream' site approximately 100 m downstream of the confluence of the glacier's two rivers: (1) a river flowing from a subglacial portal and (2) a stream flowing from a supraglacial channel. (It was not possible to access the river at any point further upstream from the sampling point due to a deeply incised ice channel and partially collapsed glacier caves.) At the same frequency, we also took samples at a gauging station we installed in the melt river approximately 1000 m downstream from the upstream sampling point. Using measurements taken at the gauging station, we derived hourly discharge measurements of the bulk melt river.

We also took samples of two groundwater springs (GW1 and GW2) within the glacier forefield every 2-5 days during the summers of 2021 and 2022. Their methane concentrations, along with the isotopic composition of the methane, were published previously in Kleber et al. (2024). Periodically, we measured the groundwater outflow rates from the two main groundwater springs. In addition, we took repeated measurements of methane ebullition, or bubbling, from vents within groundwater pools (pools formed at the site of groundwater springs). A detailed map of the field site is provided in Fig. 1 and includes the upstream melt river sampling point and gauging station, and the location of the groundwater springs.

In addition, to constrain the isotopic composition of methane in the melt river, we revisited the site and took eight water samples from the melt river as close to the subglacial portal as possible during July 2023. Using measurements from a gauging station, we also derived hourly discharge measurements of the bulk melt river during this period. It is important to note that Vallåkrabreen was surging during the summer of 2023 and the terminus was advancing up to 10 meters per day. This caused erratic discharge patterns through the drainage system, however we have assumed that the stable carbon isotopic signatures of the methane within the drainage is still relevant for our discussion.

## 2.2 Sampling and laboratory analysis

### 2.2.1 Measurement of aqueous methane in river and groundwater

Samples were taken for the measurement of aqueous methane concentrations by submerging 20 mL glass vials directly into the turbulent, well-mixed stream and capping with a gas-tight crimped cap. To prevent microbial activity during storage, samples were fixed within 24 hours with 1 mL of 1 M NaOH, then stored upside-down in the dark at approximately 4°C until analysis. The measurement of water methane concentration was conducted using the headspace method as described in Kleber et al. (2023), on a gas chromatograph fitted with a flame ionization detector (GC-FID, Agilent Technologies UK Ltd., South Queensferry, UK) at the Queen Mary University of London. Methane concentration measurements were within an analytical error of 5.5%, calculated as two-times the standard deviation of repeat measurements of 100 ppm standards ($n = 12$), and a lower detection limit of 18 nM.

The stable carbon isotopic signatures of methane ($\delta^{13}C$-$CH_4$) in the groundwater samples were analysed at the University of Cambridge in the LASER-ENVI facility using a cavity ringdown spectrometer (Picarro G2201-I, Picarro Inc., Santa Clara, California, U.S.A.) with an analytical error of 0.1‰. Samples were first diluted to create sufficient volumes for analysis by injecting 2 mL of headspace from the equilibrated sample vials into a 100 mL gas tight vial previously flushed with pure $N_2$ gas. The $\delta^{13}C$-$CH_4$ of melt river samples taken in 2023 were measured on samples that were fixed with NaOH immediately after sampling to avoid any isotopic fractionation due to microbial processes during storage. Isotopic signatures were determined by applying a cryo-focussing technique as described in (Schloemer et al., 2016) with a GC-Combustion interface II/III coupled to a Thermo Fisher Scientific MAT 253 at the Federal Institute for Geosciences and Natural Resources,

Hannover, Germany. The $\delta^{13}C$ of methane in air was measured daily as a performance test with an average standard deviation of ±0.7‰. All $\delta^{13}C$-$CH_4$ values have been reported relative to the Vienna Pee Dee Belemnite (VPDB) standard (Coplen, 2011).

### 2.2.2 Groundwater outflow measurements

Discharge measurements of the groundwater outflows were made using the float method (Turnipseed and Sauer, 2010) periodically throughout the summer and are within an error of 18% based on the standard deviation of repeat measurements

($n = 10$). A small wooden float was placed on the surface of the stream and repeat measurements of its velocity were made along a pre-determined length of stream. Depth and width measurements of this section of stream were made at regular increments and the average dimensions were used to calculate a discharge ($Q$):

$$Q = \frac{d \times w \times l}{t} \times 0.85 \tag{1}$$

where $l$ is the length of the section of stream, $d$ and $w$ are the average depth and width of this section, respectively, $t$ is the

average time the float used to travel the length and 0.85 is the coefficient used to reduce the surface velocity to average velocity.

### 2.2.3 Hydrological monitoring of the melt river

Total discharge from the main melt river was calculated using hourly stage measurements made by a Druck CS420 pressure transducer at the downstream gauging station from 03 July to 15 September 2021. The rate of discharge was measured periodically using a fluorescent Rhodamine WT dye tracing method and a Turner Designs CYCLOPS-7F fluorometer, as

described by Wilson et al. (1986). The fluorometer was calibrated after each dye tracing by measuring its voltage output in solutions made from a known volume of river water and incremental additions of dye while constantly stirring to maintain suspended sediment loads comparable to the river. Hourly stage measurements were converted to hourly discharge rates using a calibration curve (Supplementary Fig. S.1) calculated with 8 discrete discharge measurements. The calculated melt river discharge had an error of 17%, based on the greatest difference between 8 discharge measurements through dye tracing and

the corresponding discharge calculated by the calibration curve.

### 2.2.4 Ebullition measurements

Measurements of methane emissions via ebullition were made using a bespoke bubble trap as described in Walter et al. (2006), which funnelled all gas bubbles released from an individual vent into a plastic bottle of known volume (Supplementary Fig. S.2). Repeat measurements were made periodically on five discrete vents throughout the summer. Five mL of the collected

gas were extracted from the bottle and injected into a pre-evacuated 3 mL Exetainer vial. Methane concentrations and the carbon isotopic composition of methane of ebullition samples were analysed at the University of Cambridge in the LASER-ENVI facility using a cavity ringdown spectrometer (Picarro G2201-I, Picarro Inc., Santa Clara, California, U.S.A.). Samples were diluted in $N_2$ gas as described above in order to achieve adequate volumes for analysis.

### 2.3 Potential emission calculations

We calculated potential methane emissions from groundwater outflows and the melt river, which represent the amount of excess methane transported to the proglacial area by these systems. Potential emissions were calculated using a mass balance approach (Hodson et al., 2019), described in its basic form in Equation 2. For a single glacial input that equilibrates with the atmosphere before discharge into the sea, the mass balance-defined emission flux, $F_{atm}$, is

$$F_{atm} = (C_{in} - C_a) \times Q_{in},\tag{2}$$

where $Q_{in}$ is the discharge input to the proglacial area with methane concentration $C_{in}$ less the atmospheric equilibrium concentration ($C_a$, ~4 nM for fresh water at 0°C with an atmosphere of 1.8 ppm methane). The calculated potential emissions assumed that all methane above the atmospheric equilibrium concentration is degassed to the atmosphere. The likelihood of any consumption of the methane by microbial oxidation before it can be released to the atmosphere has been addressed in additional calculations.

### 2.3.1 Potential melt season methane emissions from groundwater

The equation used to calculate potential methane emissions from groundwater spring $x$, $F_x$ (kg hr$^{-1}$), is as follows:

$$F_x = \left(C_{x,CH_4} - 4\right) \times a \times Q_x \times 10^{-6},\tag{3}$$

where $C_{x,CH_4}$ is the average concentration of methane (nM) in the outflow of groundwater spring $x$, cited from Kleber et al. (2024). Conversion factor, $a$, is used to convert methane concentration from nM to mg L$^{-1}$ (1.6 x 10$^{-5}$), and $Q_x$ is the average

hourly discharge rate of the outflow (L hr$^{-1}$). $F_x$, calculated in kg hr$^{-1}$, is obtained by converting the methane concentration to kg L$^{-1}$, which is done by multiplying the whole equation by 10$^{-6}$. Hourly fluxes were extrapolated across the five melt season months.

### 2.3.2 Calculation of extent of methanotrophy in the groundwater

Using a closed-system Rayleigh function (Equation 4) (Whiticar, 1999), we calculated the percent of methane possibly lost due to methanotrophy in the GW1 spring:

$$\delta^{13}C_{CH_4,t} = \delta^{13}C_{CH_4,i} + \varepsilon \ln(1 - F),\tag{4}$$

where $\delta^{13}C_{CH_4,t}$ is the carbon isotope ratio of the methane remaining in the stream at time $t$, $\delta^{13}C_{CH_4,i}$ is the carbon isotope ratio of the initial methane in the stream outflow prior to oxidation, $\varepsilon$ is the magnitude of the carbon isotope fractionation

during methane oxidation between the outflow of the stream and $t$, and $F$ is the fraction of methane lost during this time.

### 2.3.3 Potential melt season methane emissions from the melt river

Potential melt season methane emissions from the melt river, $F_{riv}$ (kg a$^{-1}$) were calculated on an hourly basis and summed using the following equation:

$$F_{riv} = \sum_{i=1}^{n}\left(C_{i,CH_4} - 4\right) \times a \times Q_i \times 10^3 \times 3600 \times 10^6, \tag{5}$$

where $C_{i,CH_4}$ is the hourly concentration of the river at the upstream sampling point (nM) determined by linear interpolation between measured samples, $a$ is a conversion factor to convert methane concentration from nM to mg L$^{-1}$ (1.6 x 10$^{-5}$), and $Q_i$ is the discharge rate of the river measured hourly (m$^3$ s$^{-1}$) which is converted to hourly discharge in L hr$^{-1}$ by multiplying by $10^3$ and further multiplying by 3600. The discharge rate was derived by stage measurements at hour $i$, and $n$ represents the number of hours in the summer. $F_{riv}$, calculated in kg summer$^{-1}$, was obtained by converting the methane concentration to kg

L$^{-1}$, which was done by multiplying the whole equation by $10^6$.

The Lognbreen river (highlighted in yellow in Fig. 1b), which is fed from a small valley glacier to the east of Vallåkrabreen, enters the Vallåkrabreen river upstream of the gauging station. Therefore, the contribution of the Lognbreen river was removed from the total discharge measurements for methane emission calculations. Periodic discharge measurements of the Lognbreen

river were made by salt dilutions (Turnipseed and Sauer, 2010) and compared to the total Vallåkrabreen discharge rate at the corresponding time. The percent contribution from the Lognbreen river averaged 10% ($n = 4$) and thus the discharge rates of the Vallåkrabreen river were reduced by 10% for the calculation of methane emissions. The gauging station was also downstream from the confluence of the groundwater springs measured in this study, however, their overall discharge rate consistently equated to <0.01% of the total discharge rate of the Vallåkrabreen river and thus their contribution was considered

negligible.

Discharge rates of the river outside of the gauging period (26 May-02 July and 16 September-06 October) were estimated daily using the relationship between mean daily air temperature measured at a weather station 10 km from Vallåkrabreen (Sveagruva, seklima.met.no) and the sum of hourly discharge per day determined throughout the sampling period (Supplementary Fig.

S.3). The start (26 May) and end (06 October) of the melt season were selected based on the commencement of continuous plus-degree days and the commencement of continuous minus-degree days, respectively, using data from the Sveagruva weather station. Methane concentrations in the river at the start of the melt season and before the sampling period began (26 May-08 July) were estimated conservatively as the concentration of the first sample taken on 08 July (3172 nM). The concentrations in the river at the end of the melt season and after the sampling period finished (24 September-06 October)

were estimated as the average concentration of the last two samples taken at the upstream sampling point on 15 and 24 September (481 nM).

# 3 Results

## 3.1 Potential methane emissions from the melt river

Methane concentrations in the main melt river were measured at the upstream sampling point as well as at the downstream
gauging station and are plotted in Fig. 2. The upstream melt river started with high concentrations (up to 3170 nM on 8 July,
or Day of Year 189) towards the beginning of the melt season and declined to average values of ~500 nM by the end of July.
The downstream samples also began the season with high concentrations (1000 nM on 03 July, or Day of Year 184) and
declined to an average of ~440 nM for the remainder of the summer.

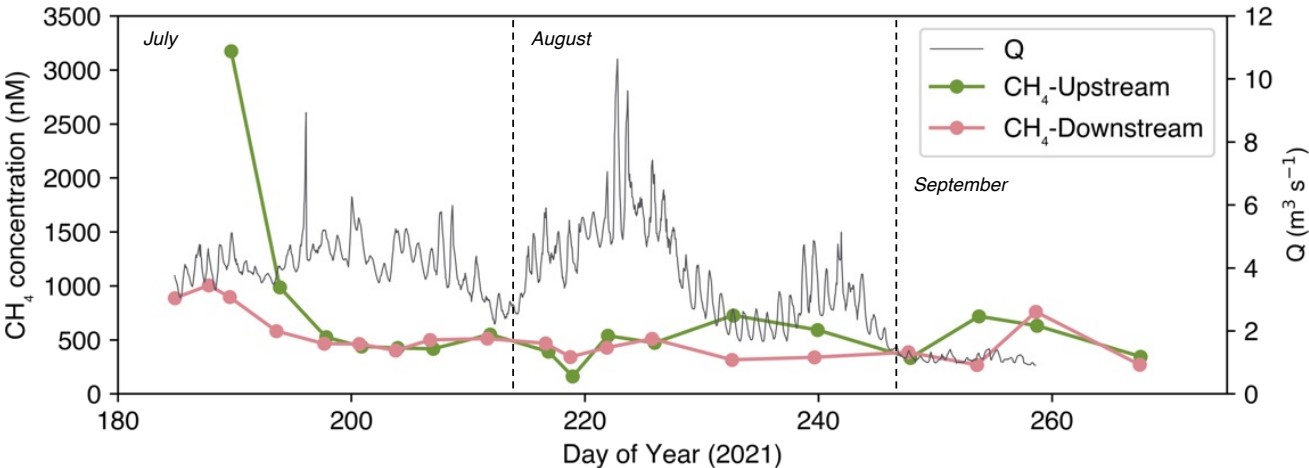

**Figure 2. Measured methane concentrations (nM) of the upstream and downstream melt river plotted over the seasonal hydrograph,**
**which provides hourly discharge measurements (Q) in m³ s⁻¹.**

Using the upstream methane concentrations throughout the summer and the discharge rates from the hydrograph, we calculated
the total amount of methane that was transported with the melt river from beneath the glacier and the corresponding potential
emissions. During the monitoring period in 2021 (between 03 July and 23 September, 82 days), approximately 274 kg (217-
342 kg) of methane were transported from the glacier margin in the melt river, which equated to an overall average of 3.34 kg
per day. This amount did not account for methane transported during the early melt season, prior to the monitoring period, or
the end of the melt season, after the monitoring period. Inferring early (26 May-02 July) and late season (24 September-06
October) discharge rates through temperature-discharge correlations suggested that the methane flux throughout June and early
October could have added an additional 345 kg (269-426 kg) of methane. Therefore, the total amount of methane transported
by the melt river during the full melt season was 618 kg (486-768 kg). After considering the atmospheric equilibrial
concentration of ~4 nM methane that remains in the water, this translated to potential methane emissions of 616 kg methane
(484-766 kg). Normalized across the glacier area (~20 km²), the potential flux was 0.23 mg CH₄ m⁻² d⁻¹.

### 3.1.1 Carbon isotopic composition of methane in the melt river

The carbon isotopic signatures of methane ($\delta^{13}$C-CH$_4$) in samples taken at the subglacial portal of the melt river in 2023 ranged from -46.5‰ to -45.3‰. The $\delta^{13}$C-CH$_4$ are plotted in Figure 3, along with the measured discharge (Q) and methane concentrations during the melt season of 2023.

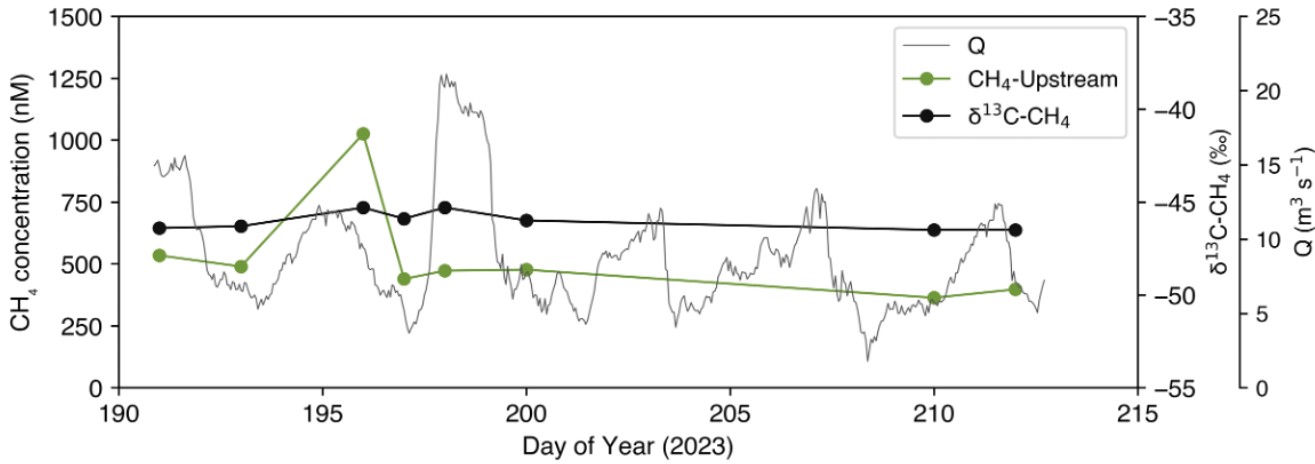

**Figure 3. Measured carbon isotopic composition of methane ($\delta^{13}$C-CH$_4$) and methane concentration of upstream melt river samples taken in July 2023 plotted over the seasonal hydrograph, which provides hourly discharge measurements (Q) in m$^3$ s$^{-1}$.**

### 3.1.2 Downstream transect of the melt river

Samples were taken in a transect along the length of the melt river in August 2022 to examine the loss of methane from the river and identify additional sources of methane to the river. Samples started at the two rivers feeding the melt river (subglacial and supraglacial) and finished at the fjord. The methane concentrations along this transect are shown in Fig. 4.

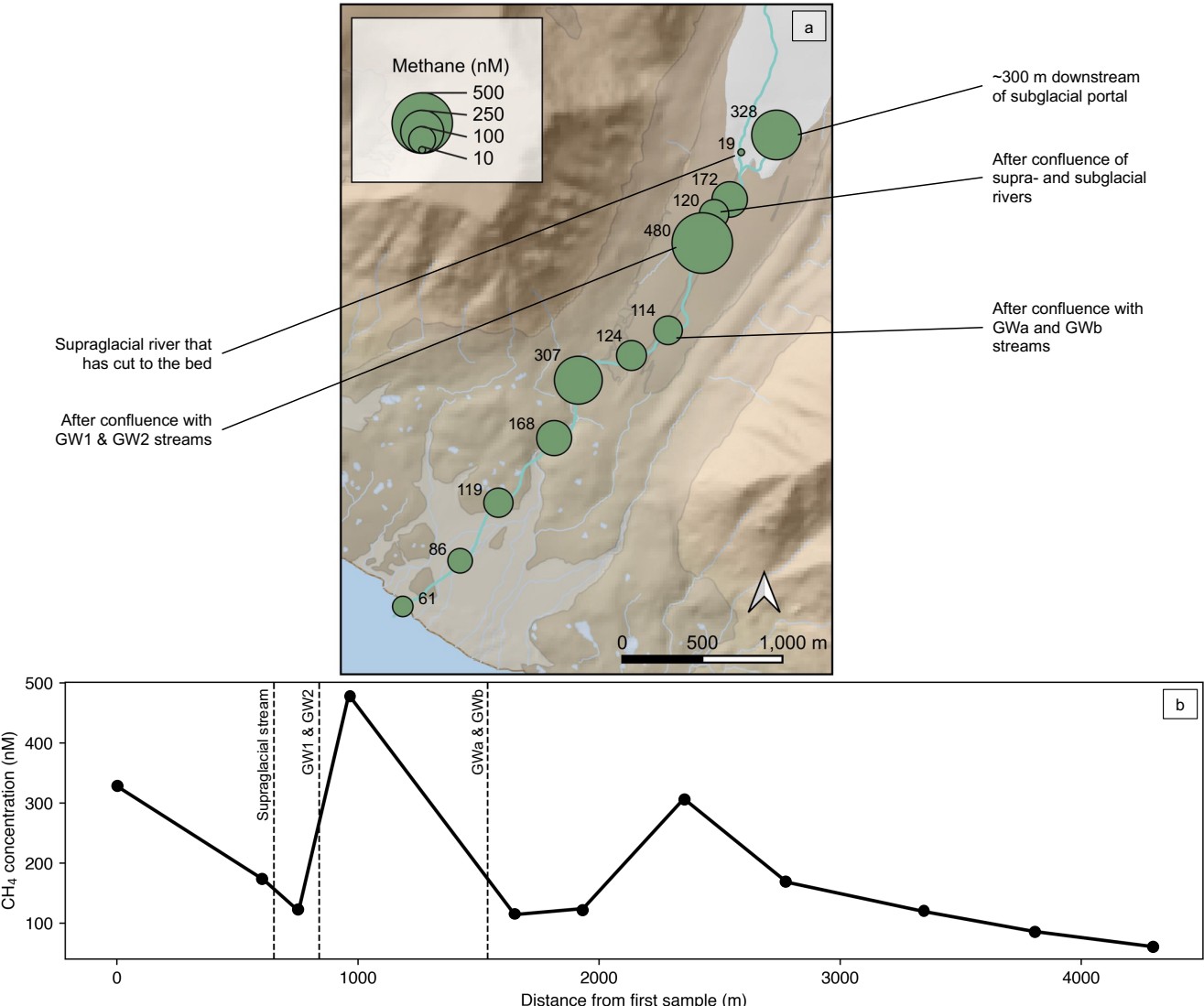

**Figure 4. (a) Transect of methane concentrations (nM) of the melt river taken on 27 August 2022. Bubble size is correlated with methane concentration, according to the legend. (b) Methane concentrations (nM) plotted against distance river has flowed from the first sampling point. Vertical dashed lines represent confluence points with other streams.**

Additional downstream transect sampling was done in 2023 to examine the changes in the carbon isotopic composition of the methane along the length of the melt river. Methane concentrations and $\delta^{13}C$-$CH_4$ measured in 2023 are plotted in Fig. 5.

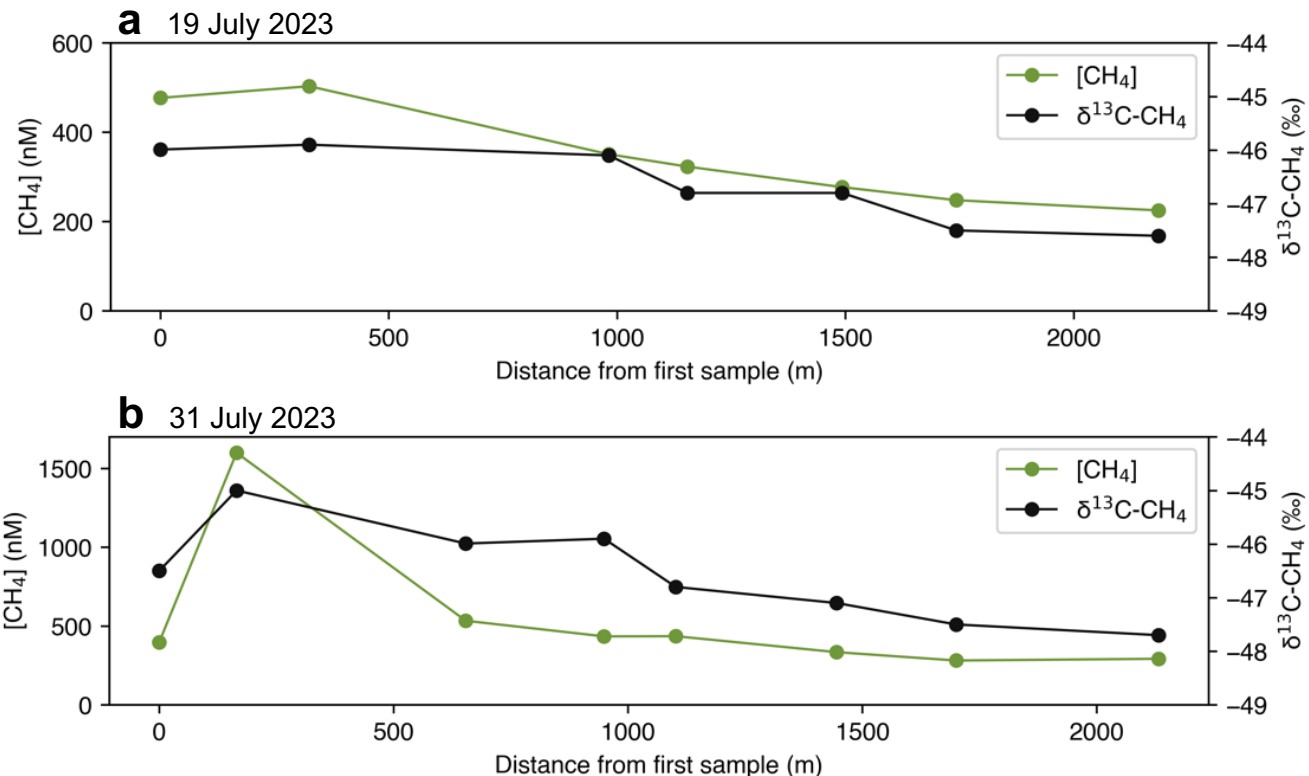

Figure 5. Transects of methane concentrations (nM) and $\delta^{13}$C-CH$_4$ of the melt river taken on (a) 19 July 2023 and (b) 31 July 2023.

### 3.2 Potential methane emissions from GW1 and GW2

Average discharge rates for GW1 and GW2 were used to calculate potential emissions. The outflow rates of the GW1 and GW2 springs were largely constant throughout the summers and averaged 1.1 L s$^{-1}$ ($\pm$ 0.19 L s$^{-1}$) and 0.86 L s$^{-1}$ ($\pm$ 0.18 L s$^{-1}$), respectively. Methane concentrations previously published for the GW1 and GW2 springs (Kleber et al., 2024), plotted in Fig. 6, were used to calculate potential methane emissions from the two groundwater sources. Seeing as the methane concentration of the GW1 spring did not fluctuate much over the course of the melt season, potential methane emissions from the GW1 spring were calculated using the average concentration over the season and equated to 244 kg methane (99-150 kg). On the other hand, the methane concentrations of the GW2 spring varied considerably over the season. Therefore, linear interpolation between the measured methane concentrations throughout the summer was used to estimate methane emissions from the spring and yielded 115 kg CH$_4$ (70.8-164 kg) throughout the melt season.

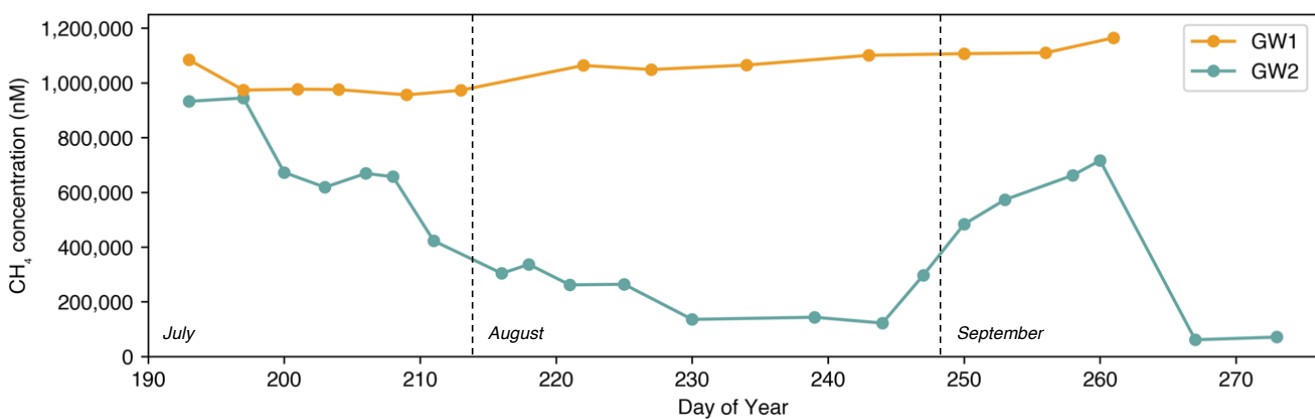

**Figure 6. Methane concentrations of the GW1 and GW2 springs as reported in Kleber et al. (2024). The GW1 and GW2 measurements were taken in 2022 and 2021, respectively.**

### 3.2.1 Transects of GW1 and GW2 outflows

Measurements of methane concentrations were made in downstream transects along the outflow streams of the two groundwater springs. The transect of methane concentrations along the outflow of the GW1 spring showed an average decrease in methane of 78% ($n = 2$ transects) within 25 m downstream of the spring outlet (Fig. 7a). The $\delta^{13}$C-CH$_4$ in the GW1 outlet stream became more enriched in $^{13}$C as the methane concentration decreased across the sampled transects (Fig. 7b). In both GW1 transects, the $\delta^{13}$C-CH$_4$ started at -44.1‰ at the stream outflow and became progressively enriched to -42.5‰ and -42.9‰ for each transect respectively at a point 25 m downstream.

The results from the GW2 spring outflow showed an average decrease in methane of 29% ($n = 6$ transects) from the outflow of the GW2 spring to where the stream met a melt river, approximately 6 m downstream (Fig. 7c). In contrast with the GW1 spring, the $\delta^{13}$C-CH$_4$ of the GW2 outlet stream showed no significant change or trend across the sampled transects and ranged from -44.6‰ to -42.5‰ (Fig. 7d).

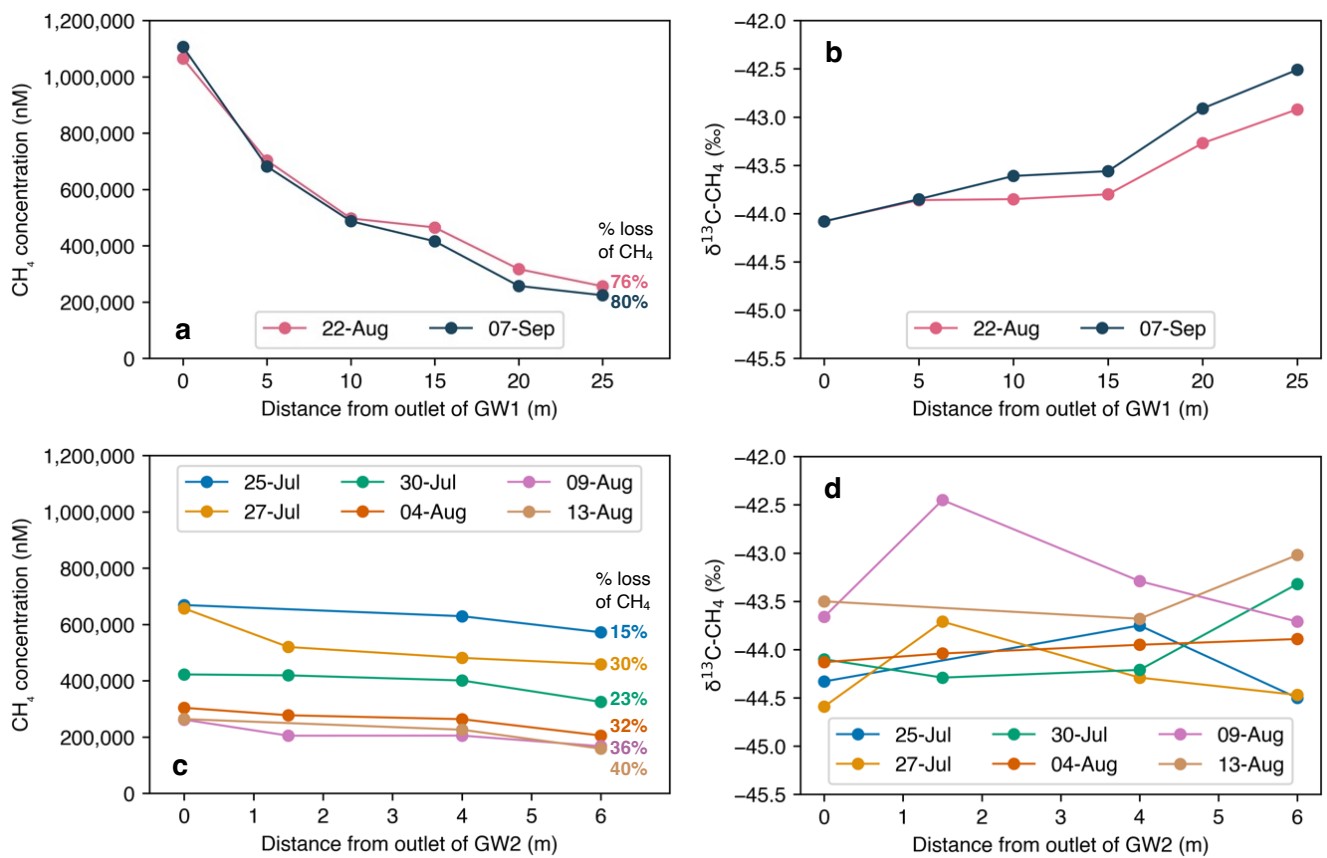

**Figure 7. (a) Decrease of methane concentrations (nM) along the outflow of the GW1 spring, including the percent loss of methane across the full length of the transect. (b) Change in stable carbon isotopic composition of methane ($\delta^{13}$C-CH$_4$) along the outflow of the GW1 spring. (c) Decrease of methane concentrations (nM) along the outflow of the GW2 spring, including the percent loss of methane across the full length of the transect. (d) Change in $\delta^{13}$C-CH$_4$ along the outflow of the GW2 spring.**

### 3.2.3 Extent of methanotrophy in GW1

The progressive enrichment of the carbon isotopic composition of the GW1 outflow stream indicated that microbial oxidation of methane, or methanotrophy, may be occurring. The magnitudes of carbon isotopic fractionation ($\varepsilon$) typically measured during methanotrophy range from 5 to 31 (Whiticar, 1999). By inserting the change in carbon isotopic composition over the length of the GW1 transect and the range of $\varepsilon$ values associated to methanotrophy into Equation 4, we calculated the potential loss of methane due to methanotrophy. We found that methanotrophy could reduce the total initial amount of methane in the stream by a maximum of 26.1% on 27 August and 36.9% on 07 September 2022 (Fig. 8). Therefore, methanotrophy could only account for a small portion of the observed methane loss over the length of the transect—anywhere from 5.0 to 34% on 27 August and 6.5 to 46% on 07 September.

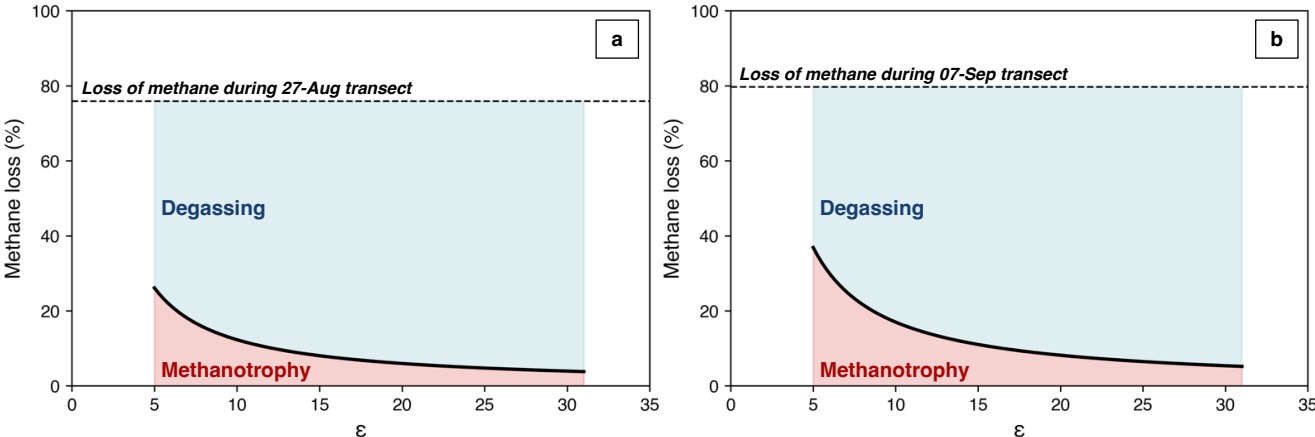

**Figure 8. Range of expected methane loss due to methanotrophy on (a) 27 August 2022 and (b) 7 September 2022. Percent methane loss is calculated with Equation 4, using the measured carbon isotopic compositions for $\delta^{13}C_{CH_{4,t}}$ and $\delta^{13}C_{CH_{4,i}}$ from the overall transect and using the range of magnitudes of isotope fractionation ($\varepsilon$) typically measured during methanotrophy (5-31). The range of methane loss due to methanotrophy is shaded pink and the balance due to degassing is shaded blue. The measured percent of methane lost across the whole transect is indicated by the dashed line in each plot.**

As it was not possible to calculate the actual contribution of methanotrophy to the reduction of methane in the GW1 outflow with our available data, we had to estimate a likely rate of methanotrophy to calculate potential methane emissions. Heilweil et al. (2016) used gas-tracer experiments to determine the relative contributions of degassing and in-stream oxidation to methane-loss from small streams, which yielded a degassing/methanotrophy ratio of 6:1. Applying this ratio to our system suggested that 14% of the methane lost across the length of the transect was lost due to methanotrophy, which fit conservatively within the range described in Fig. 8. Therefore, assuming 14% of the methane in the GW1 spring is microbially oxidized within the water column of the outflow stream, actual methane emissions due to degassing from the GW1 spring were likely to be 210 kg CH₄ (125-289 kg) across the melt season.

### 3.2.3 Spatial variability in groundwater methane concentrations

Spatial sampling of groundwaters throughout the glacial forefield was undertaken during the summer of 2021. Waters collected from 14 additional groundwater springs located between the glacier margin and the fjord revealed more springs that were super-saturated with methane (Fig. 9). Methane concentrations of the other groundwaters ranged from below the detection limit (<18 nM) to 73,300 nM, with the two highest concentration springs at 25,200 nM and 73,300 nM and hereafter referred to as GWa and GWb, respectively.

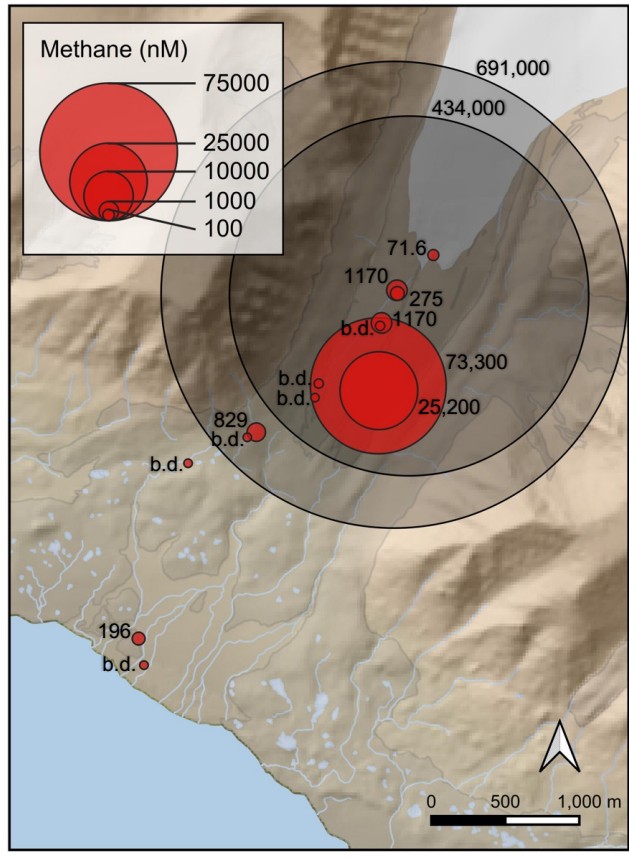

**Figure 9. Methane concentrations (nM) of groundwater springs in the forefield of Vallåkrabreen. Bubble size is proportional to methane concentration. Red bubbles represent springs sampled during one-time spatial sampling, while black bubbles represent the average methane concentrations of the GW1 and GW2 springs (Kleber et al., 2024), with bubble size extrapolated on the same bubble size scale. Concentrations below the detection limit of 18 nM are indicated by b.d.**

### 3.3 Methane ebullition

Ebullition from vents at the bed of pools formed at the outlets of groundwater springs and the bed of the groundwater streams was observed nearly constantly throughout the summer field seasons. During the summer of 2021, approximately ten vents were observed across five groundwater pools that regularly released small plumes of bubbles. Average ebullition rates measured from five of these vents ranged from 1.2 to 3.9 L hr$^{-1}$ (average: 2.6 L hr$^{-1}$). It was not possible to measure the release rates of most of the visible vents due to water that was too shallow for the bubble trap. Analysis of the gas, which readily ignited with a lighter when collected in the field, revealed an average methane concentration of 350,000 ppm (ranging from 254,000 to 482,000 ppm, $n = 9$) and an average carbon isotopic signature of -45.4‰ (ranging from -46.1‰ to -43.7‰, $n = 9$). Therefore, assuming an average release rate of 2.6 L hr$^{-1}$ of gas from 10 vents at any given time throughout the summer, total ebullition emissions were estimated as 24.0 kg CH$_4$ (20.9-27.1 kg) over the five-month melt season.

## 4 Discussion

The potential methane emissions from each hydrological system of the Vallåkrabreen catchment (melt river and groundwater) have been derived using a simple mass balance approach. This approach describes the amount of methane exceeding the atmospheric equilibrium concentration that is transported to the proglacial area and therefore has the potential to be released to the atmosphere. We used this method to avoid the large uncertainties that can be attributed to small ambiguities in predicting a gas transfer velocity (or k-value) (Wanninkhof, 1992). The dynamic nature of the glacial melt river meant that the characteristics used to predict a k-value, including discharge, channel geometry and velocity (Raymond et al., 2012) varied greatly throughout the season. Furthermore, frequent additional sources of methane to the river, as observed in Fig. 4, precluded the use of mass balance to define a k-value for the main river. In the following sections, we discuss the accuracy of the potential emissions and consider the likelihood of the removal of any methane by microbial oxidation in each of the hydrological systems.

### 4.1 Methane emissions from the melt river

We have estimated the potential emissions from the melt river to be 616 kg methane (484-766 kg) over the melt season. We suspect that this methane has largely remained unaltered by biological processes within the river and thus the microbial oxidation of methane was negligible compared to physical loss driven by the river's turbulence. This is a reasonable assumption based on the $\delta^{13}$C-CH$_4$ measured along downstream transects of the melt river (Fig. 5). The $\delta^{13}$C of the methane remaining in the river became slightly more negative over the length of the river—the opposite to what would be expected if the methane was being partially microbially oxidized along the flowpath. This assumption is in line with previous studies (e.g. Bussmann, 2013; Lilley et al., 1996; Rovelli et al., 2022), where methane is found to degas rapidly from turbulent rivers. Methane is a poorly soluble gas and its transfer velocity has been found to increase exponentially with turbulence (Herlina and Jirka, 2008). We observed rapid losses of methane from the Vallåkrabreen river (Fig. 4), where 76% of the methane was lost within a stretch of 650 m.

We used linear interpolation of methane concentrations between discrete samples to estimate potential fluxes between sampling points. This creates some uncertainty around our potential seasonal emission calculation. However, we did not find a correlation between our measured methane concentrations and the corresponding river discharge volume, and therefore could not use a discharge-weighted concentration to interpolate. This is not surprising, as we only had 17 methane concentration measurements throughout the season, which was not enough to establish a relationship between concentration and discharge volume. There are many factors that contribute to the discharge volume of a glacial river, such as snow cover extent, subglacial and englacial storage capacities, subglacial drainage configuration, glacial thermal regime and meteorological forcing. The drainage system of a glacier evolves substantially throughout the course of a melt season, thus varying the importance of each of these factors (Hodgkins, 2001; Hodson et al., 2005), and in turn varying the mechanisms of mobilization and dilution of

methane in the melt water. With this in mind, we concluded that linear interpolation was the best available method for interpolating our measured methane concentrations.

Nevertheless, the calculated potential emissions are conservative for a variety of reasons, most notably because they do not account for the methane that was degassed prior to the upstream sampling point, such as within low pressure channels at the glacier bed (e.g. Christiansen and Jørgensen, 2018). In addition, samples for the measurement of methane concentration were taken at a point approximately 50 m downstream of where the main melt river emerged from a glacial cave at the start of summer. The cave, however, gradually collapsed throughout the season, and thus the point where the river emerged from the cave moved upstream by several hundred meters by the end of the summer. The sampling point was kept constant, which meant that as the summer progressed, the river had more contact time with the atmosphere before the samples were taken. Therefore, considerable amounts of methane could have been lost from the river through degassing before the methane concentrations were measured at the sampling point. This may have had a significant effect on the calculated emission rate, yielding a lower value than reality.

Furthermore, we have based our emission calculation on methane concentrations of discrete samples taken from the melt river shortly after peak flow for the day, which was typically around 17:00. Other studies have linked methane concentrations to diurnal changes in meltwater volume, with lower concentrations expected during the daily peak flow due to dilution from larger volumes of supraglacial sources (Lamarche-Gagnon et al., 2019). Therefore, since our samples were taken around the daily peak flow period, they may represent the lowest, most diluted methane concentrations of the sampling days. This means that our linear interpolation of methane concentrations between discrete sampling times could have led to an underestimation of methane flux, as they do not account for the potentially higher methane concentrations during the daily low-flow periods.

High methane concentrations in the melt river at the beginning of summer were likely due to an accumulation of methane beneath the glacier during winter, which was then transported out along the drainage system as the river began to flow at the start of the melt season. It is important to note that due to the difficulty of accessing the site during the onset of the melt season, the earliest samples of this study were taken more than a month after the river would have started flowing. Therefore, the river likely had considerably higher methane concentrations during May and June before the sampling period of this study began, and there may have been a substantial amount of early season methane not captured in our emission estimate.

Regardless, our estimated emission rate from the glacial river is substantial considering the size of Vallåkrabreen (~20 km$^2$). When normalized across the glacier area, the flux from the glacial river equated to 0.23 mg CH$_4$ m$^{-2}$ d$^{-1}$. The Leverett Glacier, a ~600 km$^2$ outlet glacier of the Greenland Ice Sheet, has been estimated to transport between 2.78 and 6.28 t of methane from its subglacial catchment to the glacier margin over an entire melt season (Lamarche-Gagnon et al., 2019), which equates to 0.038-0.085 mg CH$_4$ m$^{-2}$ d$^{-1}$ when normalized to glacier size. Therefore, the drainage system of Vallåkrabreen, a relatively

small valley glacier, has the capacity to mobilize a larger amount of subglacial methane per glacier area than more sizeable ice sheets. This is likely due to the fundamentally different source of methane at Vallåkrabreen—largely geologic methane flushed from the rocks, as opposed to microbially produced at Leverett (Lamarche-Gagnon et al., 2019)—which is addressed later in the discussion.

### 4.1.1 Downstream transect of the melt river

The concentrations of methane at the downstream sampling site of the melt river were not always lower than the upstream values (Fig. 2), as would have been expected from continued degassing of methane from the river as it flowed downstream. This was likely the result of additional methane sources, such as groundwater streams, which entered the river along its flowpath. Figure 4 shows a transect of dissolved methane concentrations measured along the melt river, which started as close as possible to where the main river emerged from the glacier cave (about 300 m downstream) and finished where the river met the fjord (about 4 km downstream). Considerable increases in methane concentrations at points along the stream suggested several additional sources of methane to the river, some of which were not accounted for in our calculations. These continual additions of methane prevent the river from reaching atmospheric equilibrium (~4 nM) before entering the fjord. However, it has been reasonably assumed that the amount of methane discharged from the glacier terminus was largely degassed prior to reaching the fjord, but subsequently replaced with additional methane entering the river downstream.

### 4.2 Methane emissions from groundwater outflow streams

### 4.2.1 GW1 spring

Downstream transect samples of the GW1 outflow indicated rapid loss of methane from the water after it emerged from the spring. Losses of up to 80% of the initial methane concentrations were observed within the first 25 m of the GW1 outflow stream (Fig. 7c), with corresponding changes in the carbon isotopic composition of the methane remaining in the stream (Fig. 7d). An enrichment in the heavier $^{13}C$ isotope suggested that some methane was microbially oxidized in the groundwater, where molecules containing the lighter $^{12}C$ isotope were preferentially consumed. Kinetic isotopic fractionation of methane during degassing from water is very small (Knox et al., 1992) and likely negligible at such high concentrations, whereas significant carbon isotope fractionation can occur during microbial oxidation reactions of methane, such as microbially-mediated methanotrophy (Whiticar, 1999). While the physical loss of methane via degassing from the stream was likely the primary driver of methane loss, methanotrophy was an active methane sink within the GW1 outflow, consuming some of the methane before it was lost to the atmosphere.

Therefore, the 244 kg (99-150 kg) of potential emissions calculated from the GW1 spring was too high, as it was necessary to consider the consumption of methane due to methanotrophy along the outflow stream. Although microbial oxidation was clearly active in the outflow, it was not expected that the rates were exceptionally high. The low temperature of the water

(~0°C) reduces the rate of all biological activity, including methanotrophy (Lofton et al., 2014). Furthermore, the carbon isotopic composition of the methane increased only slightly (by 1.2-1.6‰) whilst the concentration of methane in the stream decreased by nearly 80%, suggesting that methanotrophy was not the dominant process contributing to the removal of methane from the stream. Our assumption of 14% methane loss due to microbial oxidation, which was based on experiments by Heilweil et al. (2016), fit conservatively within the ranges calculated using the closed-system Raleigh function (Equation 4) in Fig. 8. Therefore, our adjusted potential emissions of 210 kg $CH_4$ (125-289 kg) across the melt season are a cautious estimate of the actual emissions.

### 4.2.2 GW2 spring

Downstream transect samples of the GW2 outflow stream showed significant losses of methane, up to 40%, over a relatively short distance (~6 m) (Fig. 7c), however, there were no significant changes observed in the carbon isotopic composition of the remaining methane along the transects (Fig. 7d). There were also no additional inputs of water into this section of the stream and therefore no chance for dilution along the flowpath. This suggested that degassing of methane to the atmosphere accounted for its rapid decline from the water rather than it being diluted or microbially oxidized within the water. The rate of methane evasion from a river due to degassing can be much faster than that of microbial oxidation (Heilweil et al., 2016; Rovelli et al., 2022), especially in smaller streams where the depth is shallow and the surface-area-to-volume ratios are large. Thus, without any isotopic evidence of methane oxidation, it was assumed that the rate of removal of methane from the water due to methanotrophy, if any, was negligible relative to the rate at which it was degassed to the atmosphere. Consequently, we believe that the potential emission rate calculated for the GW2 spring, 115 kg $CH_4$ (70.8-164 kg) throughout the melt season, was a reasonable estimate for its actual emissions.

### 4.2.3 GWa and GWb springs

Spatial sampling of other groundwaters throughout the glacial forefield revealed additional springs that were super-saturated with methane (Fig. 9). Most of the saturation levels of these groundwaters were much lower than those observed in the GW1 and GW2 springs, suggesting that most of the groundwater methane emissions within the forefield were released from localized hotspots. Two of the springs, GWa and GWb, which were located within 50 m of each other, contained substantial amounts of methane (25,100 nM and 73,300 nM, respectively). This area likely represents another emission hotspot. Assuming their discharge rates were approximately 0.97 L $s^{-1}$ (the average of GW1 and GW2 discharge rates)—a reasonable assumption considering their similar size—the two springs may have released up to 20.3 kg $CH_4$ (12.6-29.2 kg) throughout the melt season combined. There was certainly additional evasion of methane from some of the less concentrated groundwater springs, as many had levels significantly above the concentration at equilibrium with the atmosphere, however these emissions were likely to be less than a few kilograms per summer and thus comparatively negligible.

### 4.3 Methane ebullition from groundwater pools

Ebullition within the groundwater pools and outflows was observed nearly constantly. Total ebullition emissions were estimated as 24.0 kg $CH_4$ (20.9-27.1 kg) over the five months where the icing was not capping the groundwater springs (typically June through October). This is a conservative estimate, as there were likely more active vents in other groundwater pools or springs that were not observed. Only 10 vents were assumed in the calculations because only 10 vents were observed to emit bubbles on a regular basis (observed at each visit to the groundwater springs). However, many more vents were observed that emitted bubbles sporadically and these were not accounted for.

### 4.4 Total methane emission estimate

The total estimate of melt season emissions from the Vallåkrabreen catchment equated to 1.0 ton of methane ($\pm$ 0.3 ton) between June and October. Methane emissions from the glacial melt river are assessed to have contributed 63% of these emissions, while the groundwater and ebullition contributed 35% and 2%, respectively. The potential methane flux from the Vallåkrabreen catchment during the melt season is equivalent to 1.7 mg $CH_4$ $m^{-2}$ $d^{-1}$ when normalized across the area of the glacier forefield, or 0.18 mg $CH_4$ $m^{-2}$ $d^{-1}$ when normalized across the area of the entire hydrological catchment. It is difficult to compare this flux to the flux of methane from thawing permafrost on Svalbard due to the extreme heterogeneity of permafrost as well as the scarcity of published measurements. However, the seasonal flux of methane per area of the Vallåkrabreen catchment is within the range of fluxes measured from permafrost on Svalbard. The few available studies have yielded contrasting results—one has found a growing season flux of methane of 0.08 g C $m^{-2}$ (0.88 mg $CH_4$ $m^{-2}$ $d^{-1}$) (Lindroth et al., 2021), while another study has found that a chamber on wet tundra generated up to 2.0 g C $m^{-2}$ over a melt season (~22 mg $CH_4$ $m^{-2}$ $d^{-1}$) (Pirk et al., 2017). Yet another study by Müller et al. (2018) found no production of methane from (nor a significant presence of genes involved in methane production within) permafrost cores taken ~350 m away from the chamber site of the study by Pirk et al. (2017).

Our melt-season emission estimate is conservative due to various limitations in the field—most notably, the inability of sampling all groundwater springs in the catchment and the inaccessibility of the melt river directly at the glacier margin. The impracticality of identifying and measuring all groundwater springs throughout the forefield made it likely that there were methane-rich springs that were not accounted for in our emissions estimate. This was apparent in the notable increase of methane in the melt river transect from 124 to 307 nM (Fig. 4) in an area where a groundwater spring had not been identified. The inability of accessing the melt river at the glacier margin meant that methane was lost from the river before methane concentrations were measured at the upstream sampling point. The methane degassed between the margin and the sampling point (a distance ranging from ~50-200 m throughout the summer) could have been substantial considering the high turbulence of the river and the rapid loss of methane observed in the river transect in Fig. 4. This may represent a considerable amount of methane not accounted for in our emissions estimate.

## 4.5 Methane source

Through isotopic analysis and the presence of ethane and propane, the methane in the groundwater of the Vallåkrabreen catchment has previously been found to be geologically sourced (Kleber et al., 2024). However, the origin of methane in the melt river had not yet been addressed. Previous studies have suggested that microbial methane production can occur in subglacial environments due to the considerable amounts of organic carbon that can be sequestered during a glacier's advance and the presence of anoxic conditions (Boyd et al., 2010; Burns et al., 2018; Dieser et al., 2014; Stibal et al., 2012; Wadham et al., 2008). It has been widely agreed in studies of methane-emitting glacial rivers across the Arctic and sub-Arctic that subglacial methane is largely microbially produced (Burns et al., 2018; Christiansen et al., 2021; Lamarche-Gagnon et al., 2019; Pain et al., 2020). However, we have found evidence that the methane in the Vallåkrabreen melt river is geologically sourced. Its carbon isotopic signatures ranged within the thermogenic realm (Whiticar, 1999) between -46.5‰ and -45.3‰ (Fig. 3), and its ethane and propane concentrations yielded wetness levels (62-91, $n = 2$) that indicate the methane originates from oil-associated thermogenic gas (Supplementary Table. S.1) (Milkov and Etiope, 2018).

The potential for methanogenesis in the subglacial environment has been found to depend largely on the sediment type and, in turn, the character of the organic substrate and its bioavailability (O'Donnell et al., 2016; Stibal et al., 2012). The rates of methane production within various subglacial sediment types have been found to vary by orders of magnitude (Stibal et al., 2012). O'Donnell et al. (2016) compared the abundance and availability of organic carbon in sediments in basal ice from glaciers overriding different substrates. They found that the Finsterwalderbreen glacier, the only Svalbard glacier in the study, which is situated less than 60 km southwest of Vallåkrabreen, contained the least amount of bioavailable organic compounds within its basal ice—an order of magnitude lower than Joyce Glacier in Antarctica, which had overridden a lacustrine environment. The dissolved organic carbon present in the basal ice of Finsterwalderbreen is thought to be mainly derived from kerogen in the bedrock, which has limited bioreactivity (Wadham et al., 2004). Considering Vallåkrabreen is situated in a similar geological and geographical setting to Finsterwalderbreen, it is expected that the basal sediments of the Vallåkrabreen catchment would offer similarly low levels of organic substrates to stimulate microbial activity.

While some methanogenesis is potentially occurring in the subglacial environment of Vallåkrabreen, it is evident that any microbially-produced methane in the drainage system is supplemented largely by thermogenic methane sourced from the rocks over which the glacier has flowed. The physical processes related to glacial advance—such as the excavation of large depths of bedrock through glacial erosion and geological faulting induced by glacial loading—can encourage the migration of deep-seated hydrocarbons to the surface where they may be introduced to the subglacial drainage system (Patton et al., 2022; Vachon et al., 2022). Alternatively, we suggest that pressurized subglacial water may route through the fractured bedrock beneath the glacier, extracting methane along its flowpath—effectively inducing a natural 'glacial fracking' process. These mechanisms

for geologic methane mobilization appear to have the capacity to yield higher methane fluxes per area within the glacier drainage system than the drainage systems of glaciers and ice sheets whose methane is solely produced by microbes.

The mobilization of methane in the glacial catchment appears to be dependent on its various hydrological systems. Groundwater has been found to play a large role in the conveyance of subterranean methane to the surface, especially at the glacier margin where unfrozen "taliks" are revealed by retreating glaciers on Svalbard (Kleber et al., 2023, 2024). These groundwater systems are active all year, and rely on sufficient snow and glacier melt to recharge their aquifers during summer
and prevent them from freezing over during winter. In permafrost regions like Svalbard, prolonged and more extreme melt seasons may expand groundwater aquifers (Liljedahl et al., 2017), potentially disrupting further geological methane stores and leading to its mobilisation into the hydrological system. Similarly, methane that has migrated further upwards into the subglacial drainage system also appears to require an active hydrological system to mobilize it, which in this case is achieved every summer. Therefore, the exceptionally high methane concentrations in Vallåkrabreen's melt river at the beginning of the
melt season indicated that methane had been stored within the subglacial aquifer throughout the winter and was flushed out once the river started flowing. Presumably, as glacial melt rates continue to increase in a warming climate, larger volumes of melt water will flow through the subglacial environment, at least in the short term. This has the potential to increase hydraulic pressures, forcing water further and deeper into the subglacial bedrock to expand the subglacial aquifer. The expansion of these aquifers could also potentially influence the thermodynamic stability of methane hydrates that may be present beneath
some Svalbard glaciers (Betlem et al., 2019), although glacier thinning and feedbacks upon both temperature and pressure associated with hydrate collapse are not easy to predict. As further substantiation of the importance of hydrological processes for methane mobilization, we found no evidence of methane diffusion through proglacial sediments into the atmospheric boundary layer during chamber measurements throughout Vallåkrabreen's forefield (Supplementary Fig. S.5). If the methane was migrating to the surface independently of the hydrological network, we would expect to have seen diffusion at the sites
we visited. Consequently, increased melt water flow due to accelerating glacier ablation is likely to dominate future changes in methane emissions by flushing out larger amounts of methane via subglacial and groundwater flowpaths until peak water production is reached.

**5 Conclusions**

Glacial groundwater on Svalbard is known to bring deep-seated geologic methane to the surface in glacier forefields and is a
550 considerable source of methane to the region's atmosphere (Kleber et al., 2023). Methane emissions from glacial melt rivers, on the other hand, have previously not been considered on Svalbard. Our seasonal investigation into the methane dynamics of the Vallåkrabreen catchment has revealed that the glacial melt river flushes significant amounts of methane from beneath the glacier and into the pro-glacial area during the melt season. While we have identified several hotspots of exceptionally methane-super-saturated groundwater seepage throughout the forefield, the glacial melt river nevertheless accounted for nearly

two-thirds of the conservatively estimated 1.0 t of potential melt season methane emissions from the total catchment. This flux makes the Vallåkrabreen forefield as strong a methane source as wet tundra on Svalbard.

Our study has shown that the meltwater of small valley glaciers like Vallåkrabreen can mobilize a substantial amount of methane, challenging previous theories that subglacial methane is largely produced microbially in the anoxic environment beneath large ice sheets (Wadham et al., 2008). We have demonstrated an alternative methane source in the glacial environment, where ancient thermogenic methane stored in the rocks beneath glaciers is flushed out by meltwater produced by the glacier. This brings thousands of smaller valley glaciers—which lack substantial subglacial environments and previously may have been discounted for their capacity to foster subglacial methane stores—into the spotlight as potential methane emission hotspots. Vallåkrabreen is one of more than 1400 land-terminating glaciers on Svalbard (Nuth et al., 2013), many flowing over geology that is rich in organic carbon, such as shale, coal and sandstone. We suspect that emissions from methane-rich glacial rivers on Svalbard are prevalent across the archipelago and may amount to a large, seasonal source of methane to the region's atmosphere, which has previously been overlooked. Our findings suggest that methane emissions from glacial rivers is likely more widespread than previously thought, and contributions from valley and mountain glaciers across the Arctic should not be discounted.

**Data availability**

All raw data can be provided by the corresponding authors upon request.

**Supplement**

The supplement related to this article is available online at: TBD

**Author contributions**

GEK and AH designed the study. GEK and LM conducted the fieldwork. StS performed the analysis of the carbon isotopic composition of methane in the river samples. GEK performed the remaining laboratory analyses with support from MT and YZ. GEK analyzed the data and drafted the manuscript. LM, AVT, MT, StS and AH reviewed and edited the manuscript.

**Competing interests**

The authors declare that they have no conflict of interest.

**Disclaimer**

Publisher's note: Copernicus Publications remains neutral with regard to jurisdictional claims made in the text, published maps, institutional affiliations, or any other geographical representation in this paper. While Copernicus Publications makes every effort to include appropriate place names, the final responsibility lies with the authors.

**Acknowledgements**

We thank the logistics department at the University Centre in Svalbard for their support in the logistical aspects of the fieldwork.

**Financial support**

This work was supported by the HYDRO-SURGE, CLIMAGAS and METHANICE projects, all funded by the Research Council of Norway (project nos. 329174, 294764 and 326285, respectively).

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
