# Peer review of "Proglacial methane emissions driven by meltwater and groundwater flushing in a high Arctic glacial catchment"

_EGUsphere, 2024_

## Author Comment (AC1)

**Reply to RC2**

**Reviewer comment:** *Because you do the upscaling for the catchment over the meltseason and claim the Vallåkrabreen is a hotspot, you also need to compare it with other CH4 emitting sites at Svalbard as a minimum and secondly from the broader Arctic.*
**Reply:** We have compared the melt river emissions to emissions estimated for Leverett Glacier in Greenland, and we have now compared the total catchment flux on a per area basis to fluxes measured from permafrost on Svalbard.

**Reviewer comment**: *Line 11: I would suggest an alternative formulation: "Glacial groundwater is a conduit of geologic methane released in front of retreating glaciers on Svalbard…" so as not to provide the indication that CH4 is produced in the meltwater itself as you conclude yourself later.*
**Reply**: We have updated the text as suggested.

**Reviewer comment**: *Line 13: It is misleading here that you mention "two summers" as the reader gets the impression that you monitored the emission and gauging over two summers when it was actually only one summer you did these measurements. I suggest that you revise the formulation here and also in line 16 to specify that the emission estimate was only for 2021.*
**Reply**: We have updated the text as suggested.

**Reviewer comment**: *Line 17: "catchment"? Do you refer to the 20 km2 or the groundwater catchment which is maybe bigger as this would include the glacial groundwater. If possible I would normalize to the area as well, so that metric is presented up front.*
**Reply**: We felt that adding the hydrological catchment size to the abstract was too much detail for an abstract, so we have added the hydrological catchment size (42 km$^2$) to the field site description in the methods.

**Reviewer comment**: *Line 18-19: Notion of proglacial areas as hotspots. This is only valid if you compare it to CH4 emissions from other CH4 sources in Svalbard and/or other Arctic CH4 emitting hotspots, such as thermokarst or Arctic wetlands. Here it would help to have the emission rates normalized per area.*
**Reply**: We have added a flux normalized per area and also later compared this to measured methane fluxes from permafrost on Svalbard in the discussion (section 4.4). This is a difficult comparison because flux from permafrost is highly heterogeneous and thus upscaling it on a per area basis can cause significant uncertainties. Likewise, normalizing the glacier emissions per area is difficult because it is not straightforward to constrain the area that you normalize it to. There are not substantial soils for CH4 production during permafrost thaw on mountain sides and mountain tops, so should such areas be included when defining the area of the glacier catchment? In addition, the region around Vallåkrabreen is not vegetated and has little to no soils for methane production during permafrost thaw, thus, the glacier forefields in this region are likely to be the main 'hotspots' of emissions. We have decided to use the area of the glacier forefield to normalize the flux per area, as we believe this is the most appropriate way to compare it to the flux from permafrost.

**Reviewer comment**: *Line 21-22: Technically, the glacial meltwater is not the source of the methane. This must be the geologic methane as you mention in the first line. I would rather write something like: "glacial melt rivers may be growing emission points/areas of subglacial CH4 across other rapidly warming regions of the Arctic." So as not suggest CH4 is produced in the meltwater, which to date have not been shown.*

**Reply**: We have updated the text as suggested.

**Reviewer comment**: *Line 24-25: This is a claim that I guess is the backdrop for many papers on Arctic CH4, but it is overstated and should be avoided in my opinion. According to the latest global assessment of the CH4 budget in Saunois et al. 2020 (https://doi.org/10.5194/essd-12-1561-2020) in Table 6 (page 1597) of global natural CH4 emissions in 2017 above 60°N is 16 (top down) and 31 (bottom up) Tg CH4 y-1. This is 7 and 10% respectively of the global CH4 emission from natural sources (60°N to 90°S) where the southern hemisphere CH4 emission is mainly from terrestrial sources and globally tropical wetlands is by far the most important source of CH4 and will continue to be as emissions are also increasing from these regions under climate warming. I would delete the sentence and focus the introduction with the second sentence in line 25-27 changing it to something like: "In recent decades, seasonal and climatic controls on methane emissions from climate sensitive Arctic systems, such as wetlands, permafrost and geological seeps, have been observed…" In this way it becomes evident that it is really the climate sensitivity that is at the heart of your story here and not the Arctic budget which you cannot say anything about.*
**Reply**: We agree with your points and have updated the text as suggested.

**Reviewer comment**: *Line 27-31: Here your text seems to directly link the potentially rising CH4 emissions from a warming Arctic to the observed temperature increase in the Arctic. I am not a climate scientist but that much I know, is that the temperature increase in the Arctic is related to the global circulation system and not local emissions of methane. Please rewrite so as not to allude to this wrong claim. Also the paper Rantanen et al. 2022 does not mention CH4 in their paper as a cause for this rapid warming.*
**Reply**: We agree with your points and have removed the sentence from the text.

**Reviewer comment**: *Line 41-63: I think this section is nicely written and frames your story well. However, you mention that CH4 have two subglacial sources, either from overridden organic carbon or geological sources. It is not clear from the text whether your study also considers subglacial CH4 derived from organic carbon at the bed in relation to the geological methane, but you only consider geological methane. The partitioning between these two sources in the specific setting of Svalbard is highly interesting, but you do not have the measurements to provide qualified assessments of this partitioning.*
**Reply**: We have updated the last paragraph in the introduction to more clearly explain what our study addresses – including investigating the origin of the methane. We have also added carbon isotopic signatures of the melt river (obtained from samples taken in 2023) to the paper, which allow us to assess the origin. (New Figure 3 attached)

**Reviewer comment**: *Line 80-82: What was the reason to sample here, where you have a mix of the subglacial and supraglacial waters, effectively diluting the subglacial waters? I could imagine that you over the season will have differing degrees of dilution due to different meltwater volumes from the two rivers. What are your thoughts on how this water source mixing may influence your interpretation of the CH4(aq) concentration results? I assume you did not have means to partition the flows from the two rivers? You already discuss the dilution of subglacial waters in the discussion, but you could get an idea if you compared water volumes with concentrations.*
**Reply**: We have added the following sentence to the methods to explain why we chose the upstream sampling point as we did: "It was not possible to access the river at any point further upstream from the sampling point due to a deeply incised ice channel and partially collapsed glacier caves." In terms of how this mixing influenced our interpretation of the CH4

concentration results – our aim was to calculate a methane flux based on a gauging station that we erected downstream and therefore included the supraglacial water. So while we would have preferred to sample as close to the subglacial portal as possible, sampling after the confluence of these two streams allowed us to apply our 'mixed' concentration to the discharge rate we measured downstream to obtain a methane flux.

**Reviewer comment**: *Line 101-104: The described methodology is somewhat unclear. The G2201-i systems I am familiar with runs on a flow rate of 40 mL min-1. Your vials contain maximum 20 mL, but you only added 4 mL N2 headspace (according to your Nat Geo 2023 paper) so did you dilute your samples or run the sample through the inlet for a short period? 4 mL would is equivalent to 6 seconds of flow at 40 mL min-1 which is by far too little to saturate the tubes and cavity and hence the reading will not be trustworthy, so you must have done something else, but you do not tell what. It would be good to provide more details as they are not provided in your Nat Geo 2023 paper either, as others working with small sample volumes would benefit from knowing what you did. As it is written now it is impossible to replicate your method due missing information.*
**Reply**: We have added a sentence to explain our method which involves diluting samples with sufficient methane concentrations to create adequate volume for analysis on the Picarro.

**Reviewer comment**: *Line 105-108: I think it is not adequate to refer the reader to another publication for your methodology. Most of your readers are not familiar at all with the "float" method for groundwater discharge and you do not even refer the reader to the right chapter in this long report of 106 pages. You may even have had to adopt the method for your specific location and this is needed to be described. I am not suggesting an exhaustive paragraph, but at least few lines of text describing the main principle of the method referring to the right chapter in the report.*
**Reply**: We have expanded on the float method to provide more detail of our methods to the reader.

**Reviewer comment**: *Line 112: Which brand was the flourometer? I assume it is very important to calibrate the flourometer prior to these measurements and how was this achieved?*
**Reply**: We have added the brand and model of the fluorometer used and have also added the following sentence to explain the calibration of it: "The fluorometer was calibrated after each dye tracing by measuring its voltage output in solutions made from a known volume of river water and incremental additions of dye while constantly stirring to maintain suspended sediment loads comparable to the river."

**Reviewer comment**: *Line 121-123: Did you not measure the isotope composition of this ebullition methane? At least it is not mentioned, but quite useful in comparison to the dissolved CH4 you also run on this machine. The difference could possibly highlight transformation of the CH4, such as oxidation?*
**Reply**: We have added the measured isotopic composition of the ebullition methane. It is slightly more negative (-45.4‰) than the isotopic composition of the dissolved methane emerging with the groundwater springs (between -44.6‰ and -43.5‰), indicating that there has likely been some oxidation in the subsurface groundwater flow path prior to its emergence at the surface. However, we did not feel it necessary to discuss this in this paper as (1) it was addressed in our previous paper (Kleber et al., 2024, Frontiers in Earth Science) and (2) it only shows evidence of oxidation before the groundwater emerges at the surface. In this paper rather, we focus only on post-emergence oxidation, as our flux calculations are based on concentration measurements made after the groundwater has emerged at the surface.

**Reviewer comment**: *Line 129: Flux calculation. Could you not say that Ca could in fact be any point downstream of place of Cin and the flux is therefore instead the total flux between upstream of that measurement point to Cin? This is a more general formulation and you can avoid that rather uncertain assumption that Ca has to attain a value of 4 nM.*
**Reply**: Our river transect measurements have shown that there are many additional sources of methane to the river and therefore we cannot use upstream vs. downstream measurements to calculate a flux or k-value. The simple mass balance approach allows us to calculate potential emissions based on the idea that all methane above the atmospheric equilibrium value could be degassed to the atmosphere.

**Reviewer comment**: *Line 147 – 148: I am not convinced that linear interpolation is suitable here. You measure every 2-5 days, but the runoff varies over diurnal cycles. This has also been demonstrated in Lamarche-Gagnon et al. 2019 where the CH4(aq) concentration is linked to meltwater volume, with lower concentrations during high flow (dilution from supraglacial melt) and higher concentrations during low flow (less dilution). So dependent on when you sample your water during the hydrograph you can either end up overestimating or underestimating concentrations in between samplings with interpolation. If there is no relation between discharge and CH4(aq) there is no other way than linear interpolation with all the uncertainties that follow. However, did you investigate whether there was a relation, for example regressing CH4(aq) downstream and Q (Figure 2)? And if so, would it not be more correct to make a discharge-weighted concentration to interpolate between discrete samples? I acknowledge that you have relatively few discrete samples, so there may not be a clear relationship also because your concentrations are relatively stable. In the end the uncertainty is likely the same as you also do a lot of assumptions with the linear interpolation. Anyways, it would good to hear your thoughts on this and your arguments and I would suggest you add your comments on the uncertainty of the chosen approach in the discussion.*
**Reply**: This is a good point and something we have thought a lot about. We found no correlation between Q and the discrete measurements of methane concentration, so we could not use a discharge-weighted concentration to interpolate. We have added a paragraph to the discussion in section 4.1 to discuss this.

**Reviewer comment**: *Line 164-170: Here you mention the discharge and methane concentration outside the gauging period, but you do not say why the periods start and end at 26th of May and end 6th of October, respectively.*
**Reply**: We have added an explanation of how those dates were selected.

**Reviewer comment**: *Figure 2: I would add "2021" after "Day of year" to make it clear that you only estimated emissions during one year*
**Reply**: We have updated the figure as suggested.

**Reviewer comment**: *Line 173 – 178: In figure 2 you use day of year which is good, but make sure also to mention that in the text, so it is easier for the reader to follow.*
**Reply**: We now refer to Day of Year in the text.

**Reviewer comment**: *Line 176-178: I think this text is not needed as it basically is the figure caption to figure 2. So I suggest to delete.*
**Reply**: We have deleted the sentence as suggested.

**Reviewer comment**: *Line 184-194: I think it could be good to calculate the average seasonal area normalized flux from the glacier catchment (20 km2) and according to the info you provided the meltseason length is 133 days with a total potential flux of 616 kg CH4 (meltriver contribution only). This is a potential flux of 0.231 mg CH4 m-2 d-1 (616 * 1000 * 1000 mg CH4/20000000 m2/133 days). It may not say a lot, but it is only for servicing the reader so it can be compared to other CH4 emitting systems in the Arctic.*
**Reply**: We have added the calculation as suggested to the results, as well as the discussion (section 4.1) and compared it to a similarly calculated per-area flux from Leverett.

**Reviewer comment**: *Line 195-196: I think the description of this transect is too short. The results conflicts with your conceptual flux model (equation 1) as it clearly shows that CH4(aq) at the outflow in to the fjord is not at the atmospheric equilibrium (4nM) but instead elevated by one order of magnitude. This in my opinion deserves a comment.*
**Reply**: We have calculated potential emissions in this study, which we describe in the methods and discussion as representing the amount of excess methane transported to the proglacial area by these systems (groundwater and meltwater). Therefore, it is irrelevant if the river reaches atmospheric equilibrium before reaching the fjord. However, it is also worth noting that there are multiple additional sources of methane to the river, some of which are not accounted for in our calculations, and therefore the continual additions prevent the river from reaching atmospheric equilibrium. The methane discharged from the glacier terminus has likely been degassed but instead replaced with new methane entering the river downstream.

**Reviewer comment**: *Figure 5: Why do you have GW1 at the two lowest panels and not on top? It is more logic to have GW1 at the top although this is only because they have the numbers 1 and 2.*
**Reply**: We have switched the panels in the figure as suggested and also switched the order that they are presented in the preceding paragraph.

**Reviewer comment**: *Line 234-238: Perhaps mention that these numbers are not included in the upscaled emissions.*
**Reply**: The numbers are included in the upscaled emissions calculation, which is discussed in section 4.2.3.

**Reviewer comment**: *Line 246: It is not clear what you mean by "groundwater pools"? How do they look and where are they placed in the landscape? What is their formation? Rather unclear and I suggest to add a larger picture than what is shown Fig. S2, perhaps showing the proglacial landscape with these pools.*
**Reply**: We have added a better description in the field methods section of what a groundwater pool is. We've also added more pictures to the supplementary

**Reviewer comment**: *Line 253-257: I agree with this more conservative approach, but your transect measurements clearly show that you do not reach atmospheric equilibrium concentrations in the river and hence the potential flux is more of a theoretic construct, which according to your measurements would be lower than the 616 kg CH4 if you use the measurements with equation 1.*
**Reply**: We have calculated potential emissions in this study, which we describe in the methods and discussion as representing the amount of excess methane transported to the proglacial area by these systems (groundwater and meltwater). Therefore, it is irrelevant if the river reaches atmospheric equilibrium before reaching the fjord. However, it is also worth noting that there are multiple additional sources of methane to the river, some of which are not accounted for in

our calculations, and therefore the continual additions prevent the river from reaching atmospheric equilibrium. The methane discharged from the glacier terminus has likely been degassed but instead replaced with new methane entering the river downstream. We have added this to the discussion.

**Reviewer comment**: *Line 263-271: If you take your transect data into account you are never in the range of reaching the atmospheric equilibrium concentrations along the river. Hence, the mass balance assumption of 4 nM is not met and the difference between Cin and Ca (which could be Cmin) is smaller than if using 4 nM and hence the emission estimate is proportionally lower. I think this deserves a comment.*
**Reply**: See reply to above comment.

**Reviewer comment**: *Line 291-294: The comparison to Leverett is ok in the sense that glacier size maybe is not so important, but you do not mention that the sources of CH4 are fundamentally different between the two sites which are the cause for the different flux magnitude. The geological source below Vallåkrabreen leads to the proportionally higher flux per area than Leverett and presumably lower oxidation in meltwater, but here all the methane is sourced from microbial productuib where a substantial CH4 oxidation takes place in the subglacial environment and along the flow path below the ice. Hence, it is possible that the CH4 production rate may be much higher than the net CH4 emissons indicate. Also, you mention in line 264-265 that the CH4 is likely unaltered (e.g. oxidized) in the meltwater and perhaps also in the subglacial environment. Thus, Vallåkrabreen and Leverett are very different glacial CH4 emitting systems. This is an interesting reflection and moderates your argument and I think this deserves a comment here.*
**Reply**: We have added a comment to the discussion about the fundamentally different sources of methane to Vallåkrabreen vs. Leverett and how that is the likely reason for the large difference of flux per area between the two sites.

**Reviewer comment**: *Line 328-333: Technically this text belongs in Methods as you describe an approach to partition the loss to CH4 oxidation. So I suggest to move there.*
**Reply**: We have moved the text to the methods, as suggested.

**Reviewer comment**: *Line 328-333: Line 335-354: I think this is results and should be moved accordingly. I like the way you do it here and it seems reasonable, but it is a bit unclear and therefore it would be good to have this important result in the appropriate place of the manuscript. I was thinking of another approach or maybe it is the same? The total potential loss (degassing + oxidation) from GW1 could be calculated using equation 1 where Cin is CH4(aq) at the source and Ca is CH4(aq) at 25 meters multiplied by the groundwater outflow and assuming no water loss along the transect. Then you calculate F from equation 4 where δ13C-CH4,t is the isotopic value at 25 meter and δ13C-CH4,i at the groundwater spring and multiply this with the total loss. Could this not work as well?*
**Reply**: We have moved the section to the results, as suggested. The suggested calculation is an alternative but similar approach and gives the reader the same message. We believe our approach with the figure as a visual is more intuitive for the reader.

**Reviewer comment**: *Also, in this text you mention that you cannot calculate the contribution of methanotrophy to the loss (line 349), but then you mention this the 14% contribution in line 350. This is very confusing and it seems contradictory, so please revise the text to make it clearer.*

**Reply**: We could not calculate the actual contribution, but only the range. The 14% was chosen based on gas-tracer experiments by Heilweil et al. (2016). We have made this clearer in the text.

**Reviewer comment**: *Line 454-455: The statement "Our findings highlight that glacier forefields on Svalbard are hotspots for methane emissions" is much stronger if you compare emissions with other known sources of CH4 from Svalbard or the Arctic. See for example this preprint paper with CH4 emissions estimates from moist tundra in Svalbard: https://essopenarchive.org/users/551125/articles/604242-moist-moss-tundra-on-kapp-linne-svalbard-is-a-net-source-of-co2-and-ch4-to-the-atmosphere (the only I could find upfront and I am sure there are others).*
**Reply**: We have added a comparison to permafrost emissions in the discussion (section 4.4).

**Reviewer comment**: *In this paper (line 424-431) the total summer (day 160 – 284) CH4 emission is between 0.039 – 0.164 g CH4 m-2, which equals 0.3 – 1.3 mg CH4 m-2 d-1. The estimate I can calculate from your catchment is 1000 kg CH4/20 km2/133 days = 0.38 mg CH4 m-2 d-1 (this figure may be subject to revision as the flux magnitude and catchment sizes are uncertain)*
**Reply**: We have added such calculations to the manuscript.

**Reviewer comment**: *So compared to these preliminary estimates the Vallåkrabreen is as strong a CH4 source per square meter as moist tundra in Svalbard. I agree that with the number of glaciers in Svalbard potentially emitting CH4 they are important for the total CH4 emission for Svalbard, but without an upscaling including other surface types we are still in the dark as to how important it is. This is not your job to do here, but I think you have to mention the comparison, ideally supplemented with more studies of CH4 emissions from wet ecosystems in Svalbard or maybe across the Arctic.*
**Reply**: We have added this to the discussion and conclusions.

---

## Author Comment (AC3)

**Reply to RC1**

**Reviewer comment:** *My main criticism is the lack of stable isotope data on the $CH_4$ from the meltwater samples, which makes the interpretation of the $CH_4$ source(s) and so the governing mechanism(s) (see line 59) of the emissions difficult.*
**Reply:** We have added stable isotope data from melt river samples taken at the upstream station in summer of 2023 in Figure 3 and wetness ($C_1/(C_2+C_3)$) to the supplementary. We have also added stable isotope data from melt river transects taken in 2023 in Figure 5. We have updated our abstract, methods, results, discussion and conclusion with the new data accordingly.

**Reviewer comment**: *The claim that "small valley glaciers like Vallåkrabreen can be a substantial source of methane, challenging previous theories that subglacial methane is largely produced microbially in the anoxic environment beneath large ice sheets" (l. 440-441) makes little sense, since if the $CH_4$ is thermogenic (which seems likely given its high concentration and the expected low bioavailable OC content in the subglacial environment) the glacier itself (or rather its ecosystem) is not the source but rather its meltwater acts as a mobiliser/carrier, and the comparison of Vallåkrabreen and Leverett Glacier and their catchment sizes (l. 291-294) is beside the point.*
**Reply**: We have changed the text in line 440 to read: "…the meltwater of small valley glaciers like Vallåkrabreen can mobilize a substantial amount of methane…". We also believe that the comparison to Leverett and the differing catchment sizes is important, as it demonstrates the key point that small glacier catchments on Svalbard may represent notable methane sources – potentially releasing a larger amount of methane per area than large ice sheets. Existing literature has largely focused on outlet glaciers to the GrIS with the idea that subglacial methanogenesis is the main source of methane in glacierized environments. Our findings provide an alternative source that bring thousands of smaller valley glaciers into the spotlight as potential methane emission hotspots. We have made these points more explicit in the text.

**Reviewer comment**: *The true focus and novelty of the study should also be made clearer in the (last paragraph of the) introduction – at the moment it's quite drowned.*
**Reply:** We have updated the last paragraph of the introduction as suggested.

**Reviewer comment**: *l. 33 Subglacial C stores have also been estimated to be significant (Wadham et al 2019 Nat Comms) and should be mentioned.*
**Reply:** We have added this important reference.

**Reviewer comment**: *l. 41-42 Vinsova et al. (2022 Glob Biogeochem Cycles) provide an overview of Arctic subglacial OC and its potential microbial degradation and could be mentioned.*
**Reply:** We have added these references.

**Reviewer comment**: *l. 102 What was "sufficiently high $CH_4$ concentration" in this case?*
**Reply:** We have removed this statement, as we have added the carbon isotopic signatures of the methane in the river from 2023.

**Reviewer comment**: *Fig 2 Is there any correlation between Q and $CH_4$?*
**Reply:** No. We have checked for a correlation between CH4 and the Q at the time of sampling, as well as the peak Q of the day and neither yield any correlation so we did not think it was important to include.

**Reviewer comment**: *I also recommend the authors fix the inconsistencies in tense (past vs. present) and voice (passive vs. active) throughout the text for a better reading experience.*
**Reply:** Thank you for pointing this out – we have fixed the inconsistencies of the tense throughout the text.

---

## Author Response (AR2)

Dear Gabriel Singer,

Thank you for your thoughtful and thorough review of our manuscript. We have addressed your issues as outlined below.

**Issue 1**

*Prompted by a reviewer comment you have inserted lines 359-367 into your revision where you detail potential effects of misestimated concentration due to sampling during higher flow on your flux estimation. I cannot follow these explanations. Your (potential) flux estimate approach basically assumes that the complete load (!) of CH4 will make it to the atmosphere (minus whatever can be assumed to be left in the water at atmospheric equilibrium). Load is concentration x discharge. Your explanation assumes that higher discharge may dilute concentration, resulting in lower fluxes. Accounting for the simultaneously higher discharge, however, should make that dilution effect nil. Your flux estimation approach assumes flux to be proportional to load, not to concentration. Please explain and/or adapt these lines.*

The point we were trying to make in this text is that since we sampled only during the peak flow period of the day, we have only measured concentrations from peak flow times, which may be lower than during low flow times of the day. We are therefore potentially applying that lower concentration to the entire day, even when the low flow periods may have had higher methane concentrations, and therefore it is a potential under-estimation of total flux from each day. Our discharge values are on an hourly basis, and thus our flux is calculated on an hourly basis. During the low-flow hours, when concentrations are potentially higher, we are still applying this lower concentration we measured during high-flow hours. We have updated the text in lines 379-385 to make this point clearer for the reader.

**Issue 2**

*I and reviewers agree with your interpretation of the CH4 found in this study to be of largely geological origin. You essentially tell a story of glacial meltwater acting as a conduit for geological methane to the atmosphere. However, two "side results" seem particularly interesting to me: a) There is no correlation between discharge and CH4 concentration at any of the investigated sites, which is what could be expected if glacial meltwater streams of various discharge get in contact with a geological source of methane which thereby gets mobilized, b) even a dry groundwater "pool" acted as a source despite complete lack of water. I wonder if the CH4 emitted to the atmosphere in the investigated glacier forefield actually needs meltwater to reach the atmosphere. May there be similarly sized emission fluxes during the cold season, without any melt water? You claim at multiple places (particularly in the abstract and discussion) that your results point to "a large climate-sensitive source of greenhouse gas" and "a climate feedback loop driven by glacier melt" or "growing emission point for subglacial methane" (all cited text from abstract). Please explain your reasoning, and consider (i) toning some of those strong claims, as well as (ii) insertion of a paragraph in the discussion that actually puts the meltwater-associated emission fluxes measured during the melting season in this study into perspective with potential non-meltwater associated*

*fluxes in the cold season that have remained completely unmeasured. Note also that if the association of emission fluxes with meltwater does not hold, then also the comparison to larger glaciers and the claimed potential importance of many small glaciers must be put into question - and this point was already critically remarked upon in the previous review round.*

We have added a paragraph (lines 355-366) to explain why a lack of correlation between concentration and discharge volume is not as concerning as implied in your comment. This is because there are many factors that impact the mobilization and dilution of methane in a glacier drainage system, and the importance or influence of each of these can vary throughout the melt season as the drainage system evolves.

We have also removed the statement about the dry groundwater pool to avoid confusion (from the paragraph starting on line 476). It is a minor part of the study that we feel does not need to be addressed, as it seems to cause confusion. First, the dry vent that was measured was located within 80cm from the groundwater pool - the groundwater pool had shifted and changed shape throughout the summer, and therefore parts became dry that were previously submerged in water. So, it is very likely that the gas being released through the dry vent was still connected to the hydrological system. The ebullition is likely due to the pressure changes of the groundwater as it reaches the surface - lower pressure means that gas is not dissolved as easily and may degas rapidly when pressure decreases. Furthermore, we did a set of chamber measurements of the sediments across a portion of the glacier forefield and found no evidence of methane flux in any of the chambers. The chambers covered areas nearby the groundwater pools and the river and represented a variety of different sediment types. If the ebullition was not connected to the hydrological system, it would be expected that there would be more diffuse emissions of methane through the sediments, which we found no evidence of. We mention these chambers in a paragraph added at the end of the discussion (lines 568-572), as well as add their results to the Supplementary Info.

Finally, we added a paragraph to the discussion (lines 553-571) to address the relationship between the hydrological systems of the glacial catchment and their mobilization of geologic methane. We discuss how increased glacier melt may lead to the expansion of aquifers in the glacial catchment, which can drive further stores of geologic methane to the surface.

Thank you for your time in reviewing and editing our manuscript. We appreciate your constructive feedback.